# Molecular Modeling and Gene Ontology Implicate SLC35F4 and SLC35F5 as Golgi-Associated Importers of Flavin-Adenine-Dinucleotide

**DOI:** 10.3390/ijms27010512

**Published:** 2026-01-04

**Authors:** Zheyun Niu, Dongming Jiang, Daniel M. Hardy

**Affiliations:** Department of Cell Biology & Biochemistry, Texas Tech University Health Sciences Center, Lubbock, TX 79430, USA; zhniu@ttuhsc.edu (Z.N.); dojiang@ttuhsc.edu (D.J.)

**Keywords:** SLC35F4, SLC35F5, SLC35F3, FAD, thiamine, choline, molecular docking, protein structure, endoplasmic reticulum, Golgi apparatus, ERO1, QSOX1, post-translational modification, disulfide bonding, glycosylation, cancer, cerebellum

## Abstract

Solute carriers (SLCs) mediate cell- and organelle-specific import and export of nutrients and metabolites required for every biochemical process that occurs in a cell. Functional studies have ascribed activities to many human genes annotated as SLCs, but more than 100 SLCs remain orphans. Here, we applied a set of computational tools to characterize the orphan carriers SLC35F4 and SLC35F5. Phylogenetic analysis grouped SLC35F4 sister to SLC35F3, a suspected thiamine transporter, in a clade with SLC35F5, and distinct from an SLC35F6/2/1 clade. Transcriptome datasets revealed a restricted function for SLC35F4 in the cerebellum, in contrast to the more widespread distribution of SLC35F5. Gene ontology identified the Golgi apparatus as the likely residence of both transporters. Conceptual docking of 71 candidate substrates predicted high affinities of SLC35F4 (10–40 nM) and SLC35F5 (0.1–0.4 nM) for flavin adenine dinucleotide (FAD), straddling that of the known FAD transporter SLC25A32 (2–4 nM), while returning much lower affinities (by 30–fold or more) for all other tested substrates. Docking to SLC35F3 returned low affinity for both FAD and thiamine as candidate substrates. Thus, SLC35F4 and SLC35F5 but not closely related SLC35F3 likely import FAD into the Golgi apparatus, where the cofactor serves as the oxidant for disulfide-bond formation during tissue-specific, post-translational modification of secretory proteins. These findings provide strong direction for the definitive experiments yet needed to confirm the carriers’ subcellular localization, transport activities, and contributions to protein maturation and trafficking.

## 1. Introduction

Cells optimize conditions for different biochemical processes by segregating them into physically and spatially distinct compartments. Establishing such diverse conditions expands the range of chemistries available for enzyme-mediated catalysis, but in turn also requires cell- and organelle-specific sequestration of enzymes, targeting of transporters, and import/export of metabolites. Compartment-specific transporters include E.C. 7 Translocase family enzymes and the non-enzymatic solute carrier (SLC) superfamily proteins [1]. Translocases derive energy from ATP hydrolysis to catalyze active transport of proteins, ions, and metabolites. In contrast, SLCs facilitate the selective uptake and distribution of vital nutrients such as sugars, amino acids, and cofactors either by passive diffusion or by secondary active transport driven by ion gradients [1,2].

Combinations of SLCs mediate essential links between spatially distinct steps in complex cellular processes. For example, in energy metabolism, unique plasma membrane SLCs transport glucose into different cell types; in red blood cells, SLC2A1 (GLUT1) facilitates basal glucose entry required for glycolytic catabolism, whereas in insulin-sensitive cells such as skeletal muscle and adipocytes, SLC2A4 (GLUT4) mediates stimulated uptake in response to elevated blood glucose concentration [3]. Downstream, SLC16A1 (monocarboxylate transporter MCT1) exports lactate produced by anaerobic glycolysis to initiate the Cori Cycle, or SLC54A1/SLC54A2 heterodimer (mitochondrial pyruvate carrier complex MPC1/MPC2) transports pyruvate from aerobic glycolysis across the inner mitochondrial membrane to initiate catabolism by the tricarboxylic acid cycle in the mitochondrial matrix [4,5]. Then, as oxidative phosphorylation fueled by reducing equivalents from pyruvate oxidation consumes ADP and generates ATP, SLC25A4 (adenine nucleotide transporter ANT1) in the inner mitochondrial membrane exports newly synthesized ATP into the cytoplasm in exchange for import of ADP back into the matrix, thereby linking mitochondrial ATP production to cytoplasmic energy demand [6]. Thus, a coordinated series of cell- and organelle membrane-specific SLCs ensures the proper distribution of nutrients and metabolites between compartmentalized metabolic processes.

Notwithstanding the established functions of many SLC’s, nearly 30% of the 458 loci annotated as SLC genes in the human genome remain “orphans” with little or no experimental evidence for their products’ substrate specificity, cellular or subcellular distribution, or carrier properties (active vs. passive transport, directionality, etc.) [1,7]. Within the extensive SLC superfamily, the SLC35 family comprises at least 17 members that predominantly localize to the endoplasmic reticulum (ER) and Golgi apparatus, where they mediate import of nucleotide sugars and metabolic cofactors for glycosylation, sulfation, and redox balance [8]. Among the seven SLC35 subfamilies (A–G), only A–D are well-characterized. They share a 10-transmembrane-segment topology and operate in overlapping pathways, shuttling CMP-sialic acid, UDP-galactose, GDP-fucose, phosphoadenosine phosphosulfate (PAPS), and other nucleotide sugars. Their activities support protein glycosylation, glycosaminoglycan assembly, glycoprotein sulfation, and oxidative protein folding. For example, SLC35A1 imports CMP-sialic acid from the cytoplasm in exchange for CMP, enabling terminal sialylation of glycoproteins and glycolipids [9]. SLC35B2 transports PAPS into the Golgi apparatus, supplying the universal sulfate donor for proteoglycan and hormone sulfation [10]. In contrast, the functions of SLC35 subfamilies E, F, and G remain poorly understood. Bioinformatic predictions suggest that some members may function outside the ER-Golgi network. Proteomic and subcellular localization studies have shown that SLC35F1 and SLC35F6 traffic to and reside in the recycling endosome and lysosome, respectively [11]. Other members may also serve functions distinct from canonical nucleotide sugar transport, including the possibility that they facilitate import of small molecule cofactors or intermediates required for non-glycosylation-related post-translational modifications within the Golgi apparatus.

Here, we report the results of computational experiments characterizing properties of the orphan carriers SLC35F4 and SLC35F5. Our results predict that both carriers transport FAD required for protein disulfide bonding in the trans-Golgi. This finding links SLC35F4 and SLC35F5 to broader aspects of protein trafficking and functional maturation across the ER-Golgi interface, particularly in the formation of intermolecular disulfides during assembly of folded protein subunits into covalently bonded supramolecular scaffolds present in secretory granules of many cell types.

## 2. Results

### 2.1. Phylogenetic Analysis of the SLC35F Family Reveals Two Divergent Evolutionary Lineages

To gain evolutionary insight into the functional divergence of SLC35F paralogs, we first conducted phylogenetic analysis of sequences from 18 representative vertebrate species (Figure 1).

We chose an SLC35C protein for the outgroup owing to the subfamily’s clearly defined functions and basal position within the SLC35 family [12], and SLC35C2 in particular, owing to its conserved GDP-fucose transporter domain and lack of F subfamily-specific structural divergence. The rooted tree resolved the six members of the SLC35F subfamily into two major, supported lineages, one with SLC35F6 basal to sister SLC35F2 + SLC35F1, and one with SLC35F5 basal to sister SLC35F3 + SLC35F4 (81/94 nodes with bootstrap support probability ≥90%). Surprisingly, despite the choice of an evolutionarily robust outgroup, the SLC35F6/2/1 and SLC35F5/4/3 lineages did not emanate from a single node indicative of a common SLC35F ancestor, but instead diverged independently from the SLC35C2 root. In the SLC35F5/4/3 clade, genome comparisons revealed extensive shared synteny among SLC35F4 loci across vertebrate lineages, including mammals, birds, reptiles, and fish (Appendix A), but lower shared synteny for SLC35F3, including apparent absence in platypus, and notably weak shared synteny among SLC35F5 loci, including apparent absence in zebrafish.

### 2.2. SLC35F4 and SLC35F5 Are Golgi-Localized Transmembrane Proteins with Distinct Tissue-Specific Expression Profiles

We next examined membrane topology and localization of the SLC35F5/4/3 subclade, among which SLC35F3 has been proposed as a thiamine transporter [13], whereas physiological substrates of SLC35F4 and SLC35F5 remain unidentified. The TOPCONS utility (for consensus prediction of membrane protein topology and signal peptides) predicted 10 transmembrane helices and intracellular N- and C-termini topologies for both SLC35F4 and SLC35F5 (Appendix A). In addition, DeepLoc v2.0 predicted that both proteins localize primarily to the Golgi apparatus, with all other compartments scoring below threshold, supporting their classification as Golgi-resident transporters (Table 1).

To identify potential organ- or tissue-specific functions of SLC35F4 and SLC35F5, we examined mRNA expression profiles in human tissues using GTEx-derived RNA-seq data (Figure 2). The two transporters displayed distinct and largely non-overlapping expression patterns, with low abundance of SLC35F4 mRNA in nearly all tissues (fewer than five transcripts per million, TPM) but elevated expression in the brain, specifically cerebellum (approximately 10 TPM). In contrast, SLC35F5 exhibited higher abundance overall in most tissues, most notably in the adrenal gland and cultured fibroblasts (30–40 TPM), as well as more modest elevation in blood vessels, skin, ovary, and thyroid.

### 2.3. Gene Ontology Analysis Predicts Functions of SLC35F4 and SLC35F5 in Golgi-Localized Cofactor Transport Required for Post-Translational Modification

To characterize SLC35F3/4/5 subclade diversification further, we applied a structure-guided method for Gene Ontology (GO) prediction, TransFun, that combines AlphaFold3-based structural models with sequence information to generate high-confidence functional annotations (Figure 3). The analyses identified strong associations of the two transporters with multiple biological processes, confirmed their predicted functions as transmembrane transporters, and identified the Golgi apparatus as their most likely location. Corresponding GO annotation of SLC35F3 also confirmed its transporter function and a most likely Gogi localization (Figure 4). The pLDDT statistics from the AlphaFold3 models specified consistently High or Very High confidence in the structures of the carriers’ functionally important transmembrane segments (Appendix A).

GO annotation correctly identified the activity of SLC25A32, but in contrast to the SLC35F paralogs, predicted its localization to the cellular endomembrane system (Figure 4), consistent with its established function as a mitochondrial FAD importer [14,15].

To identify potential SLC35F4 and SLC35F5 substrates, we conceptually docked 71 candidate small-molecule cofactors to the proteins’ AlphaFold3-based structures (Appendix A), and visualized the top 15 candidate substrates (Figure 5).

The docking analysis identified a shared set of candidate substrates for SLC35F4 and SLC35F5, with relatively high predicted affinities, most notably FAD, but also including guanine nucleotides (GDP, GTP, cGMP) folate derivatives (Folic acid, THF, Methylene-THF, Formimino-THF), and nicotinamide cofactors (NAD^+^, NADPH). Docking analysis of SLC35F3 (Figure 6) identified an overlapping set of candidate substrates (guanine nucleotides, folate derivatives, and nicotinamide cofactors) with predicted affinities in the –8 to −10 kcal/mol range. Beyond the shared interactions, each SLC35F paralog displayed distinct substrate preferences. Notably, the top 15 candidate substrates for SLC35F3 did not include FAD, and the analysis also predicted low affinity for thiamine.

SLC35F5 bound FAD with a minimum docking energy of −13.7 kcal/mol and showed strong stability across multiple poses (Figure 5). FAD also ranked as the highest affinity substrate for SLC35F4, albeit with slightly higher docking energy (−10.9 kcal/mol). To establish context for the FAD binding results, we docked the same panel of candidate substrates to SLC25A32 (Figure 6), which showed a distinct preference for binding FAD, with a minimum docking energy of −11.8 kcal/mol, in the range of the docking results for SLC35F4 and SLC35F5. These docking energy minima correspond to FAD affinities of 10 nM for SLC35F4, 0.1 nM for SLC35F5, and 2 nM for SLC25A32. To determine if the predicted affinities of SLC35F4 and SLC35F5 for FAD truly differed from those for the other top substrate candidates, we performed pairwise post hoc analysis of affinity differences and created a heatmap illustrating how each ligand performed for the two proteins (Figure 7). FAD stood out as the only substrate exhibiting universally higher predicted binding affinity (*p* < 0.001 for all comparisons) among the 15 top candidates.

### 2.4. Structural Basis of FAD Binding by SLC35F4, SLC35F5, and the Canonical FAD Transporter SLC25A32

To determine if FAD binding to SLC35F4 and SLC35F5 shares structural features with binding to canonical FAD carriers, we compared their docking profiles to that of the mitochondrial FAD carrier SLC25A32. Visualizing the complexes with Discovery Studio identified distinct structural contributions to the high FAD docking affinities among the three proteins. In SLC35F4, FAD occupied an embedded position across the central region of the α-helix bundle, with its isoalloxazine and adenine ring systems buried in hydrophobic cavities, and its phosphate–ribose backbone extended along the binding channel to form a stable hydrogen bond (Figure 8).

Stabilizing interactions of SLC35F4 with the flavin moiety of the ligand included H-bonds between the 2-oxo group (acceptor) of the alloxazine ring system and Arg 146 (R146; donor) of the polypeptide and Arg 146 (R146; donor) of the polypeptide and between the 14′ hydroxyl (donor) of the ribityl and N126 (acceptor), andN126 (acceptor), as well as a π–donor H-bond between N-5 of the alloxazine ring and the side chain of W35. In addition, L132, L133, V130, and A185 side chains lined a hydrophobic pocket providing further stability through alkyl and π–alkyl interactions. Stabilizing interactions with the adenosine moiety included an H-bond between the purine C-6 exocyclic amino group (donor) and G249 (acceptor) of the polypeptide, as well as a π–lone pair interaction of L252 and alkyl and π–alkyl interactions of A253, I256, and L257 with the purine heterocyclic rings, with N255 and V258 further strengthening the polar network via H-bonds to the ribose group. In sum, H-bonding, π effects, and hydrophobic interactions collectively stabilized the SLC35F4–FAD complex.

In SLC35F5, FAD occupied a partially buried binding pocket extending from near the surface to the central region of the helix bundle, with its isoalloxazine and adenine rings positioned within hydrophobic cavities (Figure 9).

Stabilizing interactions of SLC35F5 with the flavin moiety of the ligand included H-bonds between the 12′-ribityl hydroxyl (donor) and S424 (acceptor) of the SLC35F5 polypeptide, between the phosphate–ribose backbone (acceptor) and L416 (donor), and between the alloxazine N-1 (acceptor) and S423 (donor), with I426, L425, and L429 side chains engaged in π–sigma and π–alkyl interactions within a compact hydrophobic layer around the aromatic ring. Stabilizing interactions of the adenosine moiety included π–π T-shaped stacking and π–lone pair with the F363 side chain, combined H-bonding and aromatic stacking with W251, and a π–sulfur interaction with the C248 sulfur atom that collectively enhanced hydrophobic and directional binding forces. Overall, the SLC35F5-FAD complex shared features with SLC35F4 (hydrophobic cavities for heterocyclic rings, polar channel for ribityl group) in the context of a multi-point binding network dominated by hydrogen bonds and supplemented by π interactions.

Similar to SLC35F4 and SLC35F5, FAD bound SLC25A32 across the core region of the helix bundle, with its phosphate–ribose backbone extending through the inner cavity and the adenine and alloxazine rings buried in hydrophobic pockets (Figure 10).

In SLC25A32, the flavin moiety lay enclosed by a tight hydrophobic layer with its aromatic system engaged in π–sigma and π–alkyl interactions with I65, P284, A283, and I87, whereas the purine moiety formed multiple carbon-hydrogen bonds with G84, G91, and L141. The six-membered purine ring engaged in a π–π T-shaped stacking interaction with W142 to form a pronounced aromatic core, and the five-membered ring formed a π–sulfur interaction with the sulfur atom of M166, further conferring directional and structural stability to ligand binding. In sum, the cooperative effects of π and hydrophobic interactions primarily stabilized the SLC25A32–FAD complex. Table 2 summarizes the interactions predicted to stabilize FAD binding to the three carriers.

## 3. Discussion

Computational methods have become indispensable tools for exploring protein functions. Indeed, in the six decades since the idea of a molecular clock [16,17] gave birth to the field of molecular evolution, the tools themselves have evolved from rudimentary, sequence-based grouping of proteins into gene families with conserved structural features [18,19], to identification of functional motifs [20] and the advent of proteomics [21], to current AI-based methods for computationally intensive tasks ranging from the conceptually simple (but practically difficult) prediction of mutations’ effects on known protein activities [22,23] to the monumental task of defining the universe of protein structure families that molecular evolution has created [24,25]. Accordingly, here we applied a suite of computational tools, beginning with sequence-based phylogenetic analysis and ending with conceptual docking studies, to discern the likely relationships among, localization of, and substrates for the orphan solute carriers SLC35F4 and SCL35F5. The results collectively point to functions of these carriers as FAD importers in the secretory pathway.

SLC35 family carriers generally mediate import of nucleotide sugars and other substrates required for glycosylation and sulfation reactions in the secretory pathway, driven by the obligatory antiport of the nucleotide monophosphate products of those reactions [1,2,26,27]. Our findings expand the breadth of SLC35 family activities to include plausible functions of SLC35F4 and SLC35F5 in the transport of FAD into the Golgi apparatus and, by extension, possibly also into the ER owing to the carriers’ obligatory upstream transit. FAD synthesis from riboflavin (vitamin B2) occurs in the cytoplasm via sequential action of riboflavin kinase (to produce flavin mononucleotide, FMN) and FAD synthetase 1 [28]. Accordingly, the cofactor must then be transported from its cytoplasmic pool into its various target organelles. In the inner mitochondrial membrane, SLC25A32 imports FAD required for normal energy and one-carbon metabolism [14,15]. In the ER and Golgi apparatus, FAD serves as a required cofactor for the formation of disulfide bonds [29,30,31,32,33,34,35,36,37] but, despite the fundamental importance of disulfide bonding in the maturation of secretory proteins, its necessary importers have remained unidentified.

Disulfide bonding begins in the rough ER, with co- and post-translational formation mostly of intramolecular disulfides catalyzed by the combined action of protein disulfide isomerase (PDI) and ER oxidase 1 (ERO1) enzymes [29,30]. PDI activities facilitate both sulfhydryl oxidation via formation of mixed PDI-substrate protein intermediates, as well as isomerization via reduction and reoxidation to correct scrambled disulfides [32,33]. This dual function of PDI explains the paradoxical observation that formation of correct disulfide bonds requires a redox buffer of reduced/oxidized glutathione (GSH/GSSG) that favors reduction: GSSG provides the potential for oxidation of vicinal cysteines in the active site of PDI that in turn oxidize sulfhydryls of substrate proteins, whereas GSH sustains an overall reducing “redox poise” necessary for reduction and isomerization of scrambled disulfides [29,30,32,33]. In parallel, the flavoenzyme ERO1 catalyzes oxidation of GSH to regenerate the GSSG that drives oxidation of PDI required for initial sulfhydryl oxidation [31,32,33]. Thus, ultimately, FAD supplies the oxidation potential for disulfide bonding in the ER. Downstream in the Golgi apparatus, Quiescin Sulfhydryl Oxidase enzymes QSOX1 and QSOX2 catalyze the direct, PDI-independent sulfhydryl oxidation that produces intermolecular disulfide bonds of higher order multimers [34,35,36]. The isomerase activity of PDI then corrects scrambled disulfides driven by a reducing GSH/GSSG buffer as in the ER. QSOX1 and QSOX2 also derive their oxidation potential from FAD. However, no studies have yet identified ER- or Golgi-associated importers that supply ERO1 and QSOX enzymes with the FAD required for their activity in intramolecular disulfide bonding and covalent multimer assembly [34,35,36], processes critical for the normal function of secretory cells and for protein homeostasis in various diseases, including cancer [38,39]. Our conceptual docking experiments to SLC35F4, SLC35F5, and SLC25A32 yielded comparable substrate binding profiles, with the highest and clearly preferential affinity of all three transporters for FAD. Accordingly, we propose that SLC35F4 and SLC35F5 evolved within the larger SLC35 solute carrier family to mediate the essential import of FAD required for disulfide bonding in the secretory pathway.

Our analyses of tissue expression data (from [40]) revealed a potentially unique function of SCL35F4 in the cerebellum, in contrast to the widespread distribution and likely action of SLC35F5 in most tissues, consistent with experimental findings of a previous study [41]. The increased abundance of SLC35F4 in cerebellum (mean of 10–11 TPM, as compared to ≤2 TPM in other tissues), suggests a specialized function in cerebellar granule cells, which comprise an overwhelming majority of cerebellar neurons. The roughly comparable cerebellar expression of SLC35F5 (mean of 12–16 TPM) shows that SLC35F4 does not serve as the sole FAD transporter in this tissue. SLC35F4 and SLC35F5 may function in distinct subpopulations of cerebellar cells that require FAD transporters with different affinities. Alternatively, SLC35 family members can form functional heterodimers [12], so association of SLC35F4 and SLC35F5 in the same cerebellar cells could bestow unique properties to Golgi-associated import of FAD. Notably, dysfunction of cerebellar neurons likely contributes to many neurological disorders [42], including pathologies influenced by reelin [43]. Brain regions involved in neurotransmitter release and synaptic plasticity are highly dependent on redox balance and cofactor supply, and accordingly, our GO analysis for SLC35F4 returned likely functions associated with “neuron projection,” “presynapse,” and “axon terminus.” Together, these findings suggest that SLC35F4, alone or in combination with SLC35F5, may confer unique FAD transport activities in support of metabolic coupling-dependent processes in the cerebellum, such as synaptic vesicle maturation, monoamine metabolism, or redox-sensitive signaling. Moreover, other expression analyses revealed decreases in the abundance of SLC35F4 in various tumor types [44], suggesting that its loss may disrupt cofactor homeostasis or Golgi-associated signaling, thereby influencing metabolic reprogramming or antioxidant responses in cancer cells. This observation further suggests that SLC35F4 may act as a tumor suppressor-like factor, loss of which could affect tumor progression.

Classical ultrastructure studies of secretory cells from a wide variety of tissues have identified unique matrix specializations in exocytotic vesicles, such as the Weibel–Palade bodies of endothelial cells [45], chromaffin granules of adrenal cells [46], acrosomes of spermatozoa [47,48,49], secretory granules of gastrointestinal goblet cells [50], and cortical granules of eggs [51]. Many such vesicles contain supramolecular scaffolds formed by assembly of relatively few protein components into high-order polymers reinforced by intermolecular disulfide bonds. These assemblies include polymers of prepro-von Willebrand Factor in Weibel–Palade bodies [52], of chromogranin A in chromaffin cells [53], of acrogranin and zonadhesin in sperm acrosomes [49,54,55,56], of the MUC2 mucin precursor in goblet cell vesicles [57], and of the reelin protein in neurons [58,59]. Secretory vesicle matrices can provide infrastructure that enables differential exocytosis of the vesicle contents, with immediate release of some soluble components but delayed release of others that require the action of proteases to digest the covalent polymers [47,48,52,53,56]. This vesicle “degranulation” process oftentimes generates bioactive polypeptide fragments, including functionally mature von Willebrand Factor [52] and MUC2 [57], as well as granulins and epithelins [49]. Thus, transport of FAD into the Golgi apparatus by SLC35F4 and SLC35F5 may support disulfide bond formation underlying multiple aspects of secretory cell biology.

The supply of FAD in the Golgi apparatus is essential not only for QSOX enzyme activity but also may be required to maintain luminal redox balance, support post-translational modifications, and ensure protein quality control [60,61]. Indeed, to ensure proper protein folding, stability, and modifications such as glycosylation and sulfation, the Golgi apparatus must not only import specific activated substrates or cofactors such as nucleotide sugars and PAPS [26,27,62,63,64] but must also maintain conditions necessary for the enzymes that use them. In this regard, formation of disulfide bonds catalyzed by QSOX1 has been shown to regulate Golgi-resident glycosyltransferase activities [65].

Molecular docking results for SLC35F4 and SLC35F5 revealed significant overlap of predicted substrate preferences, binding affinities, and stabilizing interactions with those of the mitochondrial FAD transporter SLC25A32. These carriers differ in family origin and fold but nevertheless appear to use a similar recognition mode: π and hydrophobic interactions anchor the flavin ring, while hydrogen bonds stabilize the phosphate–ribose backbone. This structural convergence suggests that SLC35F4 and SLC35F5 independently evolved FAD-binding sites functionally analogous to SLC25A32. Unfortunately, it is difficult to relate the three carriers’ predicted binding affinities (range 0.1–40 nM depending on pose) to FAD concentration because of uncertainty about the cofactor’s free intracellular concentration. One study [66] reported 220 amole of FAD per HeLa cell that, assuming a mean cell volume of 1000–3000 µm^3^ [67,68], yields a calculated top concentration of ~200 µM, whereas another study [69] reported 2–17 amol/cell in five cell lines, including HeLa, corresponding to a low concentration of 0.7 µM. The top estimate likely reflects both free and bound FAD, whereas the low may be a closer approximation of free FAD concentration. Regardless, the low value of 700 nM lies well above the 40 nM predicted affinity of SLC35F4 (from the lowest affinity pose) for FAD, as well as the predicted 4 nM predicted affinity of SLC25A32 (also from the lowest affinity pose). Thus, within the limits of our approach and estimates of intracellular FAD concentration, SLC35F4 and SLC35F5 may function as physiological FAD importers.

Our docking studies also provide insight into the antiport substrates that might drive FAD import, as they identified guanine nucleotides as candidate ligands, albeit with lower (but still physiologically relevant) predicted binding affinities. ERO1 enzymes in the ER and QSOX enzymes in the Golgi apparatus both directly re-oxidize the FADH_2_ product from sulfhydryl oxidation by reducing molecular O_2_ to form H_2_O_2_ [32]. Consequently, there is no need for antiport of the product from the reaction that consumes FAD. Nevertheless, it is possible that the GDP or GMP products of glycosylation reactions that consume nucleotide sugar substrates such as GDP-mannose and GDP-fucose serve as antiport substrates for FAD import by SLC35F4 and SLC35F5. This inference highlights a limitation of prior studies showing that ectopic expression can drive trafficking of SLC35F2–5 to the plasma membrane [70,71] and/or confer choline [70], queuosine [71], or thiamine [13] uptake activities, but nonetheless did not establish substrate specificity by examining transport of other candidate substrates, or fully account for the carriers’ typically predominant localization to the Golgi apparatus and need for an antiported, companion substrate.

The separate divergence of the SLC35F5/4/3 and the SLC35F6/2/1 clades from the root in our phylogenetic analysis rather than a single node representing a common SLC35F ancestor suggests the clades became functionally distinct early in the molecular evolution of the subfamily, presumably driven by the action of adaptive selection. Indeed, beginning with broad phylogenetic criteria for candidate selection, Burtnyak et al. recently identified SLC35F2 as a queuine/queuosine importer [71]. Also, SLC35F1 and SLC35F6 reside in the endosome/lysosome pathway [11], where their substrates have not been identified, suggesting the divergence of the SLC35F5/4/3 and SLC35F6/2/1 clades may reflect the evolution of specialized transport functions in different cellular compartments. Furthermore, structure-based (not sequence-based) classification of the entire human SLC superfamily grouped the SLC35, SLC39, and SLC57 families together in the DMT (“Drug Metabolite Transporter”) PFAM clan, with SLC35F subfamily members intermingled with members from each of the other five SLC35 subfamilies rather than clustered monophyletically as a distinct clade [72]. Remarkably, SLC35F4 and SLC35F5 grouped together as a basal branch of the SLC35/57 clade, alongside nearest neighbors from the SLC57A subfamily, whereas SLC35F3 grouped sister to SLC35A2 (a UDP-galactose carrier) in a large clade that included members of all seven SLC35 subfamilies (A–G) [72]. This observation is consistent with our docking results for SLC35F3, which remains enigmatic. Our predicted docking affinities do not support its ascribed function in thiamine transport [13] or a function in FAD transport, but also did not identify a clearly preferred substrate despite the sister phylogenetic relationship of its sequence to SLC35F4. The incongruence between our limited, sequence-based and reported, structure-based phylogenies further emphasizes the importance of accounting for the action of selection on evolution of gene family members; sequence comparisons provide information for reconstructing the ancestry of a gene overall, but further selection analysis must be done to infer structure/function relationships and identify possible drivers of their divergence (import and antiport substrate preferences, target organelles, etc.) [73,74]. Such studies might provide some insight into the processes that drove the emergence and functional evolution of SLC35F3.

Our molecular docking analyses used mostly static protein structures, so they cannot account for protein flexibility or shape changes during ligand binding. This approach provides useful structural information on potential binding modes and key interacting residues, but remains a simplified approximation of the ligand–protein recognition process. Indeed, this limitation may underlie our inability to identify a clearly preferred ligand for SLC35F3. Molecular dynamics modeling could partly overcome this shortcoming, but ultimately, any computational analysis would require validation by direct experimental measurements. Nevertheless, our studies, though less sophisticated and powerful than molecular dynamics analyses, have generated compelling and readily testable hypotheses for the biochemical and biophysical experiments that remain necessary to validate the proposed SLC–FAD interactions and understand the cellular implications of those interactions. Possible validation strategies include gain-of-function experiments for direct assessment of FAD transport activity, isothermal calorimetry or surface plasmon resonance to measure binding affinity (K_D_), and systematic, site-directed mutagenesis of key residues in the predicted binding pocket followed by functional analyses to confirm and refine the molecular interactions. Likewise, loss-of-function experiments such as gene knockouts could provide insight into effects on protein folding and trafficking, though loss of disulfide bonding capacity would likely cause catastrophic defects leading to cell death and embryo lethality. Nevertheless, conditional knockouts in a late-developing cell type such as male germ cells could be highly informative.

## 4. Materials and Methods

### 4.1. Tissue-Wide RNA Expression Analysis of SLC35F4 and SLC35F5

RNA expression levels of SLC35F4 and SLC35F5 across human tissues were retrieved from the Genotype-Tissue Expression (GTEx v8) database (https://gtexportal.org/; accessed 1 July 2025) [75]. Transcript abundance was expressed as transcripts per million (TPM), following normalization and quality control procedures described by the GTEx consortium. For each gene, expression values across tissue types were summarized as bar plots, where tissues were grouped according to GTEx definitions. TPM values represent the median across biological replicates within each tissue.

### 4.2. Transmembrane Topology and Subcellular Localization Prediction

Using full-length standard amino acid sequence of human SLC35F3, SLC35F4, SLC35F5 and SLC25A32 retrieved from the neXtProt database, we predicted the transmembrane topology of SLC35F4 and SLC35F5 by first generating a membrane insertion free energy profile using TOPCONS (http://topcons.cbr.su.se/) [76,77], then generating a consensus topology model by integrating results from six widely used prediction algorithms. We next predicted subcellular localization using DeepLoc 2.0, a neural-network-based algorithm (https://services.healthtech.dtu.dk/services/DeepLoc-2.0/, accessed on 17 December 2025) [78] trained on curated eukaryotic proteins that make predictions based on sequence-derived features and provide probabilistic assignments to cellular compartments (e.g., Golgi apparatus, mitochondria, plasma membrane).

### 4.3. Phylogenetic Analysis of the SLC35F Subfamily

Protein sequences of SLC35 family members in 18 representative vertebrate species were retrieved from the NCBI database. Multiple sequence alignment was performed using MUSCLE v5, a fast and accurate algorithm optimized for medium-to-large protein datasets [79]. The resulting alignment was subjected to maximum-likelihood tree reconstruction using IQ-TREE v2.1.3, with automatic model selection (ModelFinder) and branch support assessed via ultrafast bootstrap [80]. The tree was rooted using SLC35C2 as an outgroup, based on its conserved GDP-fucose transporter domain and phylogenetic basal position. The resulting tree was visualized and annotated using iTOL v6, incorporating clade coloring and gene annotations [81].

### 4.4. Synteny Analysis

Synteny of the SLC35F4 and SLC35F5 was analyzed using the Genomicus v93.01 AlignView browser (http://www.genomicus.biologie.ens.fr/; accessed 1 July 2025) [82]. Homo sapiens was selected as the reference species, and the analysis was rooted at Euteleostomi (~420 million years ago) to cover the major vertebrate lineages included in this study. For each locus, Genomicus provided orthologous genomic segments across representative vertebrate species.

### 4.5. Gene Ontology (GO) Term Prediction

To predict the function of SLC35F3, SLC35F4, SLC35F5 and SLC25A32, we applied TransFun (https://github.com/jianlin-cheng/TransFun, accessed on 17 December 2025) to analyze the involving biological process, molecular function and cellular component pathways. The prediction was based on both the amino acid sequence and the AlphaFold3 PDB structure of these proteins [83]. The results were then visualized using R (v4.2.2) with the ggplot2 package.

### 4.6. Ligand Docking and Pairwise Statistical Analysis

To identify potential endogenous substrates of SLC35F3, SLC35F4, and SLC35F5, we compiled a panel of 71 candidate small molecules based on their associated GO terms. We also docked the panel to SLC25A32, a known mitochondrial FAD transporter. We retrieved the four protein structure files from the AlphaFold3 database [84,85], and ligand structures from PubChem in SDF format and converted to PDBQT for docking. We then performed molecular docking using AutoDock Vina v1.2.6, which estimates ligand-protein binding affinities (ΔG, kcal/mol) based on an empirical free-energy scoring function [86,87], and visualized the 15 highest predicted binding affinities in 3D using Python 3.10.

For SLC35F4 and SLC35F5, we applied one-way ANOVA to test for differences of binding affinity among ligands, followed by Bonferroni-corrected pairwise comparisons for post hoc pairwise test, and visualized the results as heatmaps generated in R using the ggplot2 package, with red shading indicating stronger binding of the row ligand relative to the column ligand (*p* < 0.05).

### 4.7. Docking Visualization and Interaction Mapping

Because SLC35F4 and SLC35F5 exhibited uniquely high predicted affinities for FAD, we further visualized their optimal docking conformations in Discovery Studio 2025. For each protein-FAD complex, we generated two-dimensional interaction diagrams, highlighting hydrogen bonds, hydrophobic contacts, electrostatic interactions, π–π stacking, and van der Waals forces between FAD and key amino acid residues. Interaction residues were annotated directly on the diagrams, and color-coding was used to distinguish interaction types.

## 5. Conclusions

We conclude that SLC35F4 and SLC35F5 may import the FAD oxidant required for disulfide bond formation in the Golgi apparatus and possibly also the ER. Our results also suggest that the postulated thiamine transport function of SLC35F3 should be re-examined. These computational predictions provide plausible hypotheses for future, definitive experiments focused on characterizing the carriers’ subcellular localization and in vitro/in vivo transport activities, as well as their possible contributions to protein maturation and trafficking.

## Figures and Tables

**Figure 1 ijms-27-00512-f001:**
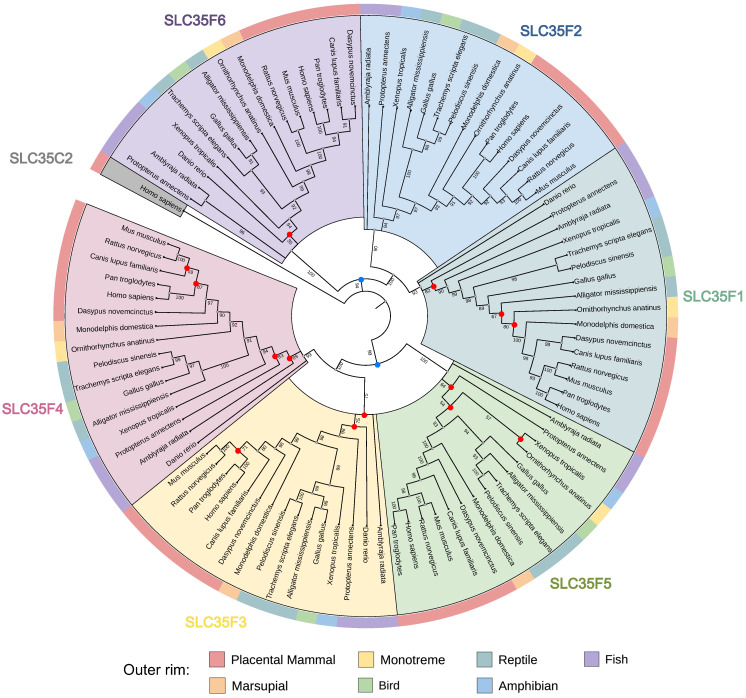
Phylogenetic analysis of the SLC35F subfamily in diverse vertebrates. Shown is a phylogram depicting relationships among SLC35F paralogs (F1–F6; shaded sectors) across major vertebrate taxa, with SLC35C2 as outgroup (shaded gray) to root the topology, and bootstrap support values (1000 replicates) for distal nodes of branches to indicate robustness of the inferred clades. Red dots mark nodes with bootstrap support <90%, and two blue dots denote the basal nodes for the SLC35F6/2/1 and SLC35F5/4/3 clades.

**Figure 2 ijms-27-00512-f002:**
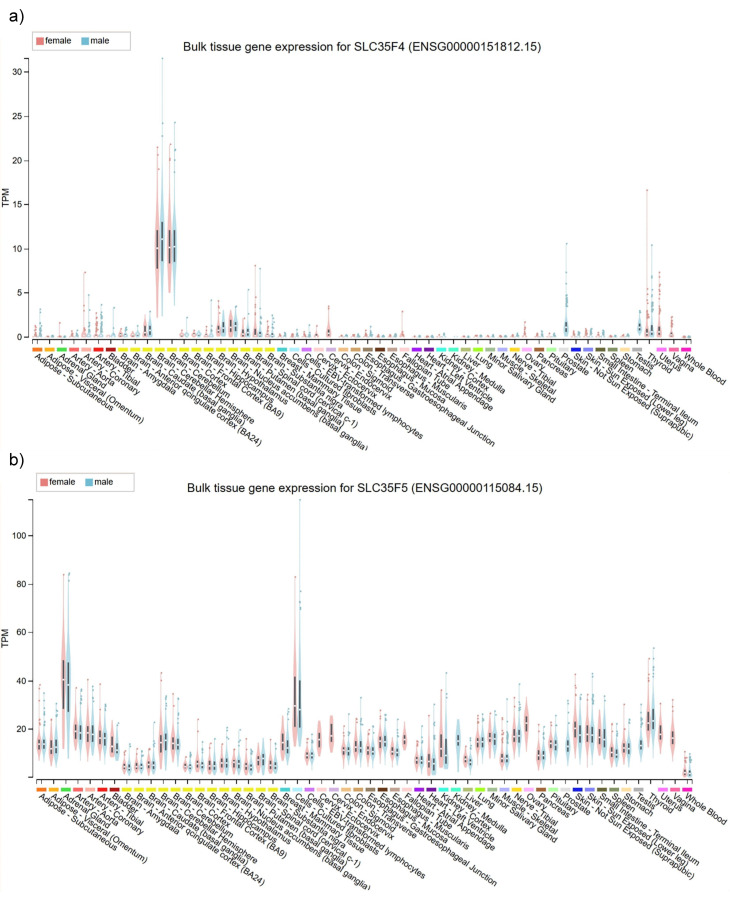
(**a**) SLC35F4 and (**b**) SLC35F5 transcript expression in human organs. Upper panel: Bar plots summarizing normalized abundance (transcripts per million, TPM; y-axis) of SLC35F4 in major human tissues (dataset’s standard tissue definitions; x-axis) using aggregated data from the GTEx project. Lower panel: Normalized abundance of SLC35F5.

**Figure 3 ijms-27-00512-f003:**
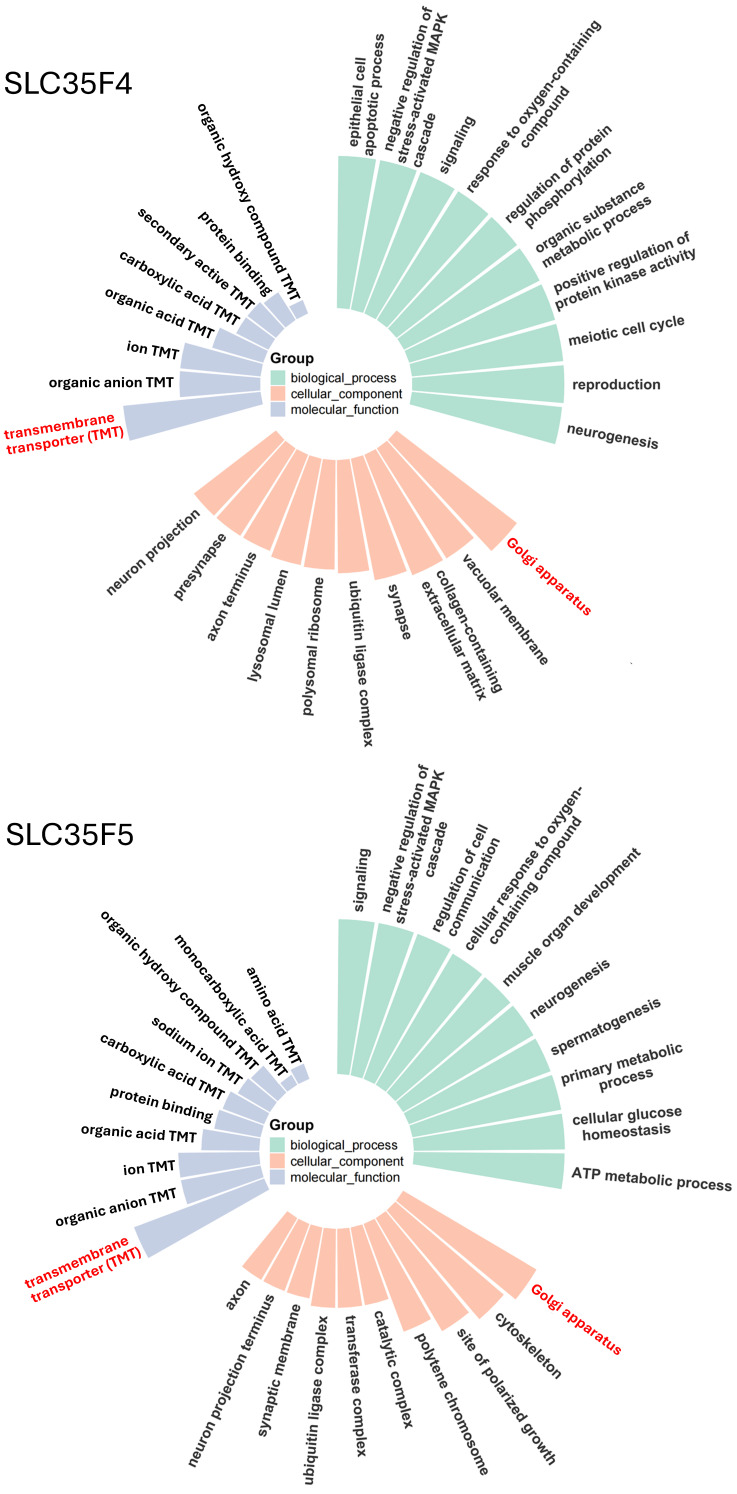
Functional annotation of SLC35F4 and SLC35F5. Upper panel: Gene Ontology (GO) annotation of SLC35F4 by Transfun. Colored segments correspond to specific GO terms in three categories (Molecular Function, Biological Process, and Cellular Component), with lengths representing the functional assignment confidence score (value range from 0 to 1). Lower panel: corresponding GO annotation of SLC35F5.

**Figure 4 ijms-27-00512-f004:**
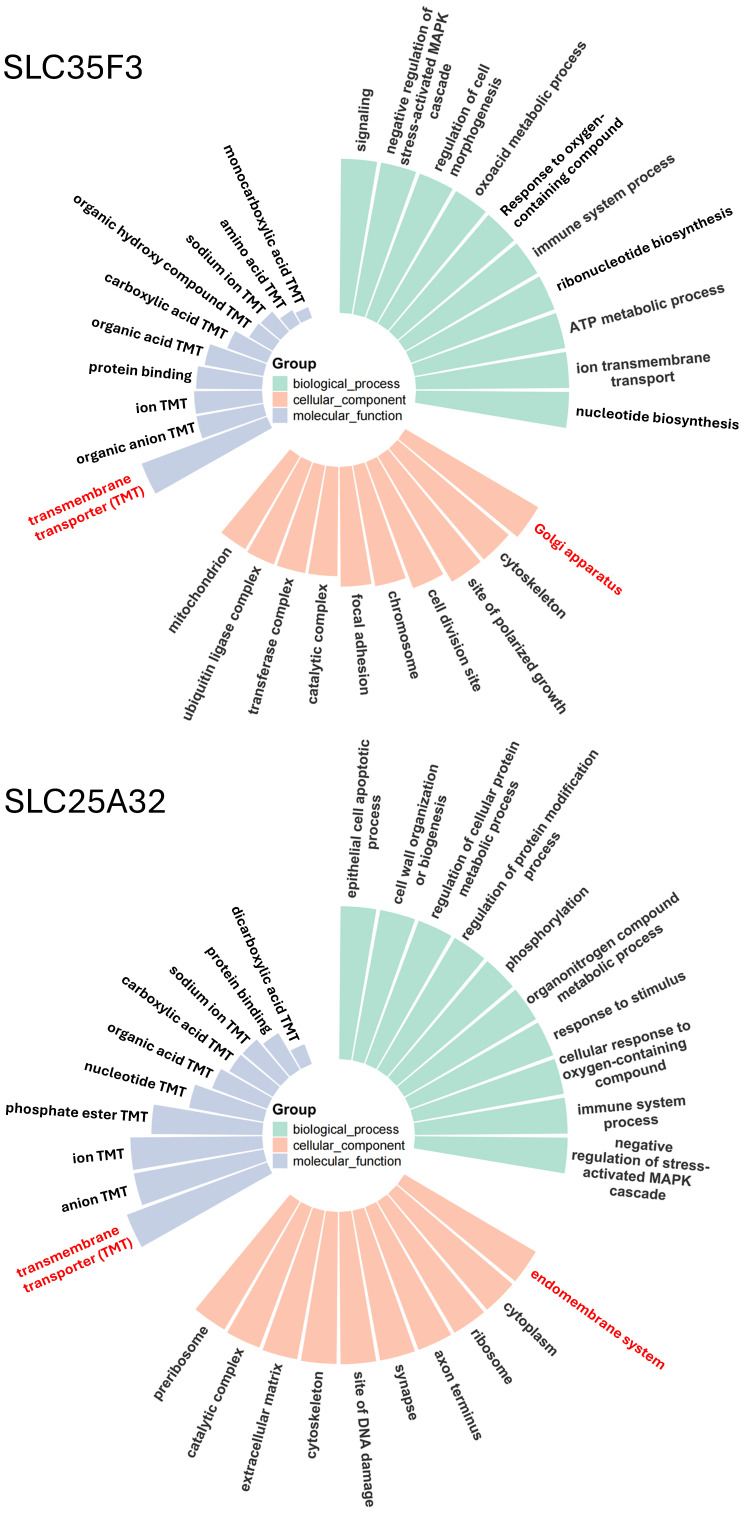
Functional annotation of SLC35F3 and SLC25A32. Upper panel: GO annotation of SLC35F3, a suspected thiamine transporter; panel features as noted in the Figure 3 legend. Lower panel: GO annotation of SLC25A32, a known mitochondrial FAD transporter [14,15].

**Figure 5 ijms-27-00512-f005:**
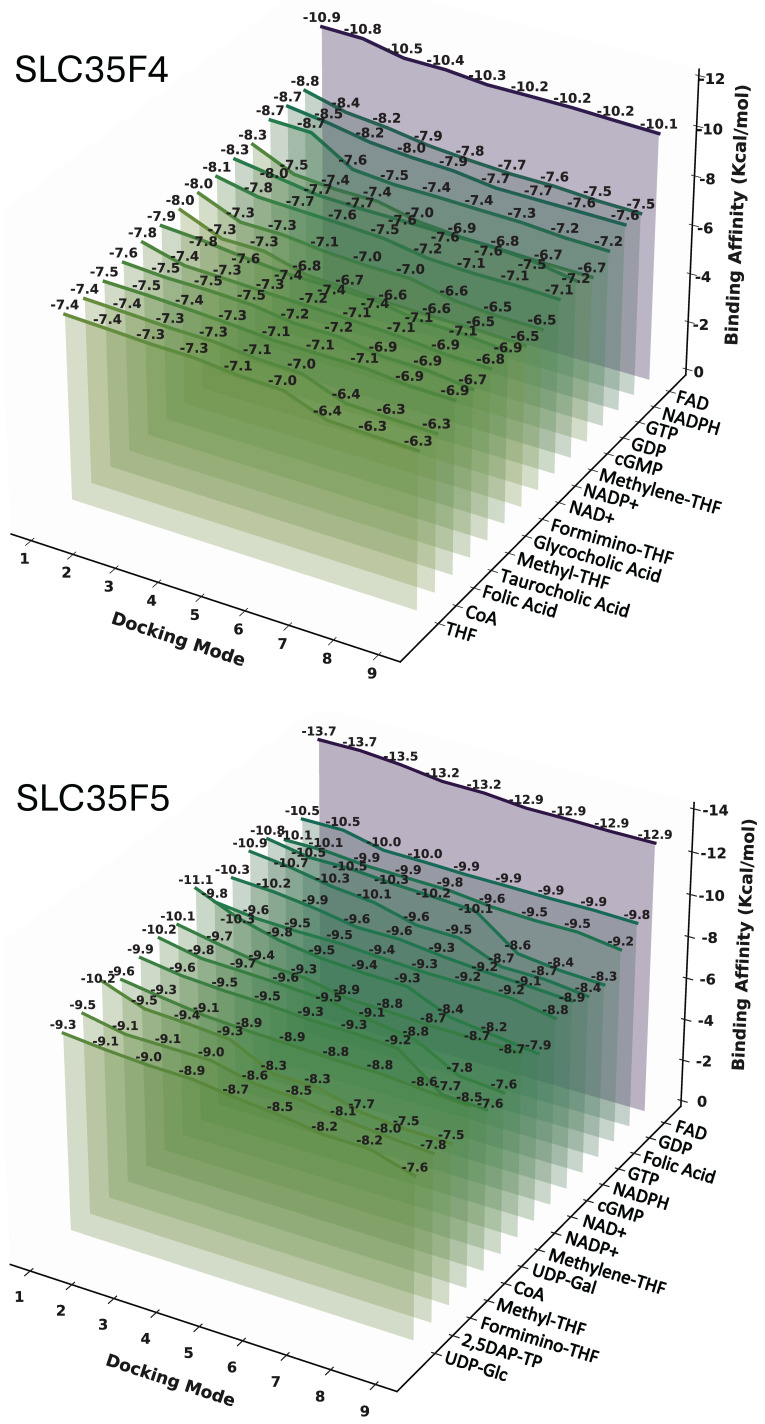
Comparative docking of candidate substrates to SLC35F4 and SLC35F5. The 3D matrix plots show binding affinities (z-axis) of the top 15 candidate substrates (y-axis) for each of nine different docking modes (x-axis). Color fade from green to purple indicates progressively stronger relative binding affinities (more negative free energy).

**Figure 6 ijms-27-00512-f006:**
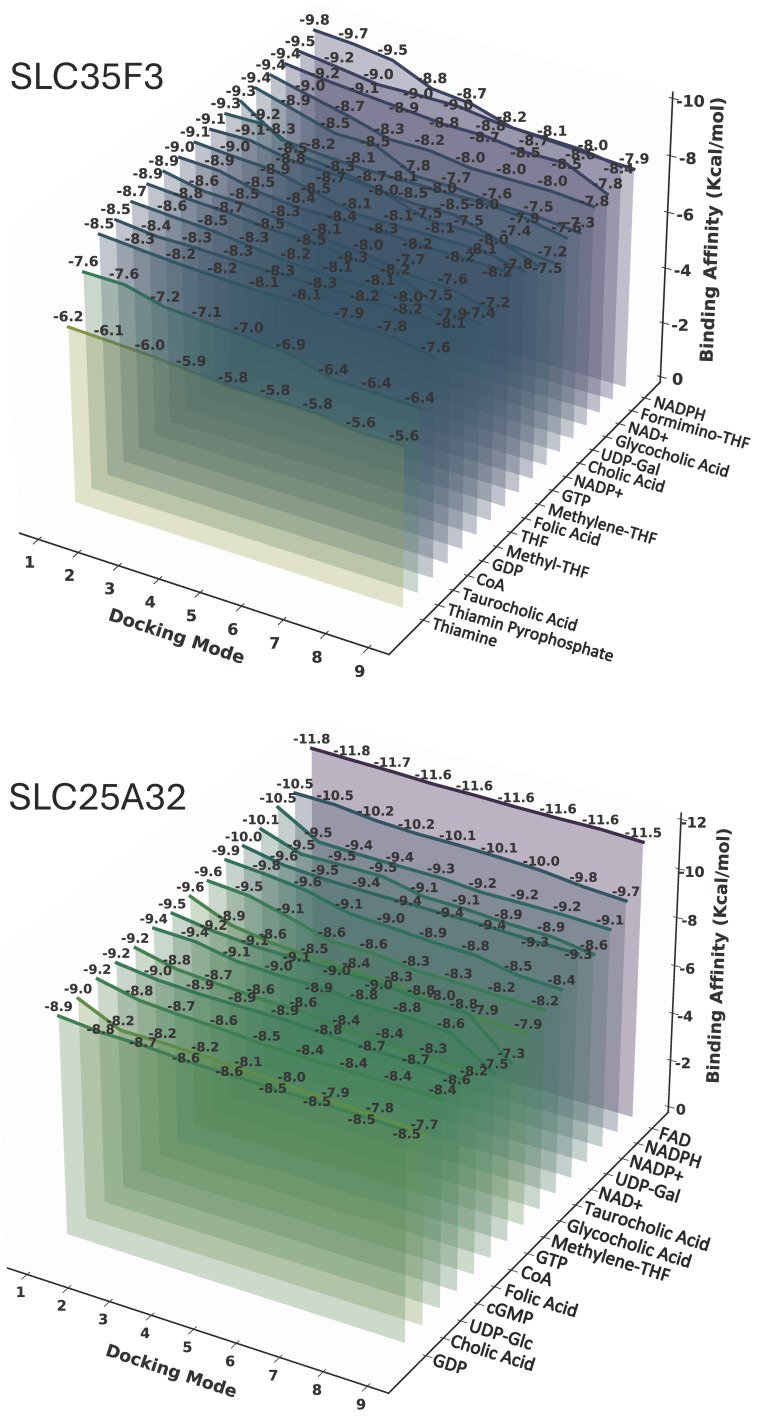
Comparative docking of candidate substrates to SLC35F3 and SLC25A32. Panel features are as noted in the legend for Figure 5.

**Figure 7 ijms-27-00512-f007:**
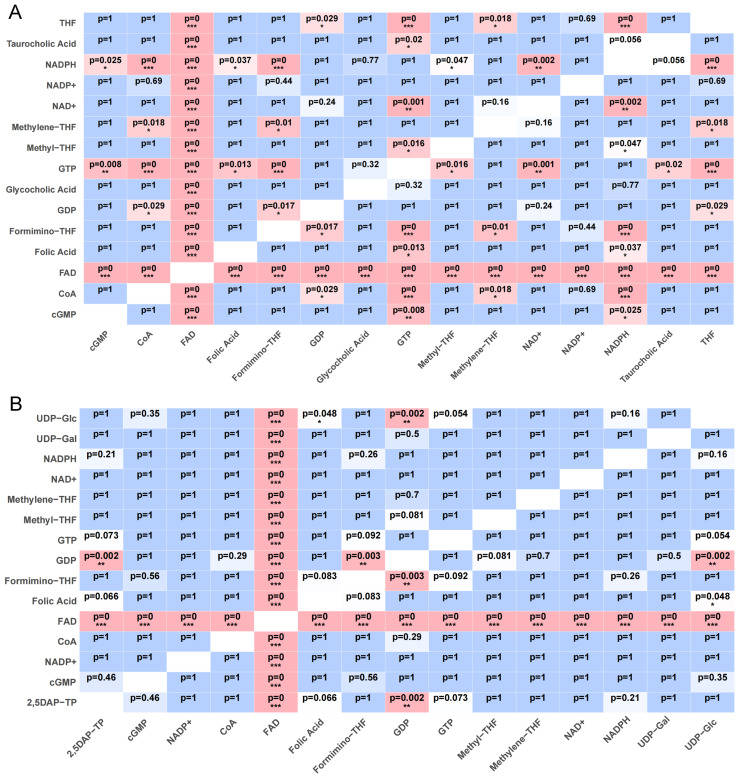
Post hoc heatmaps of docking affinity differences between the highest affinity candidate substrates for SLC35F4 and SLC35F5. (**A**) Pairwise comparisons of SLC35F4 docking scores for the 15 highest affinity substrates performed by initial one-way ANOVA across all candidate ligands, followed by Bonferroni-corrected post hoc pairwise tests. (**B**) Pairwise docking score comparisons of SLC35F5 substrates. Each tile lists the *p*-value for the row vs. column pairwise comparison; pink shading denotes *p* < 0.05, blue shading denotes *p* > 0.05, and number asterisks (*, **, and ***) denote *p* < 0.05, *p* < 0.01, and *p* < 0.001 respectively.

**Figure 8 ijms-27-00512-f008:**
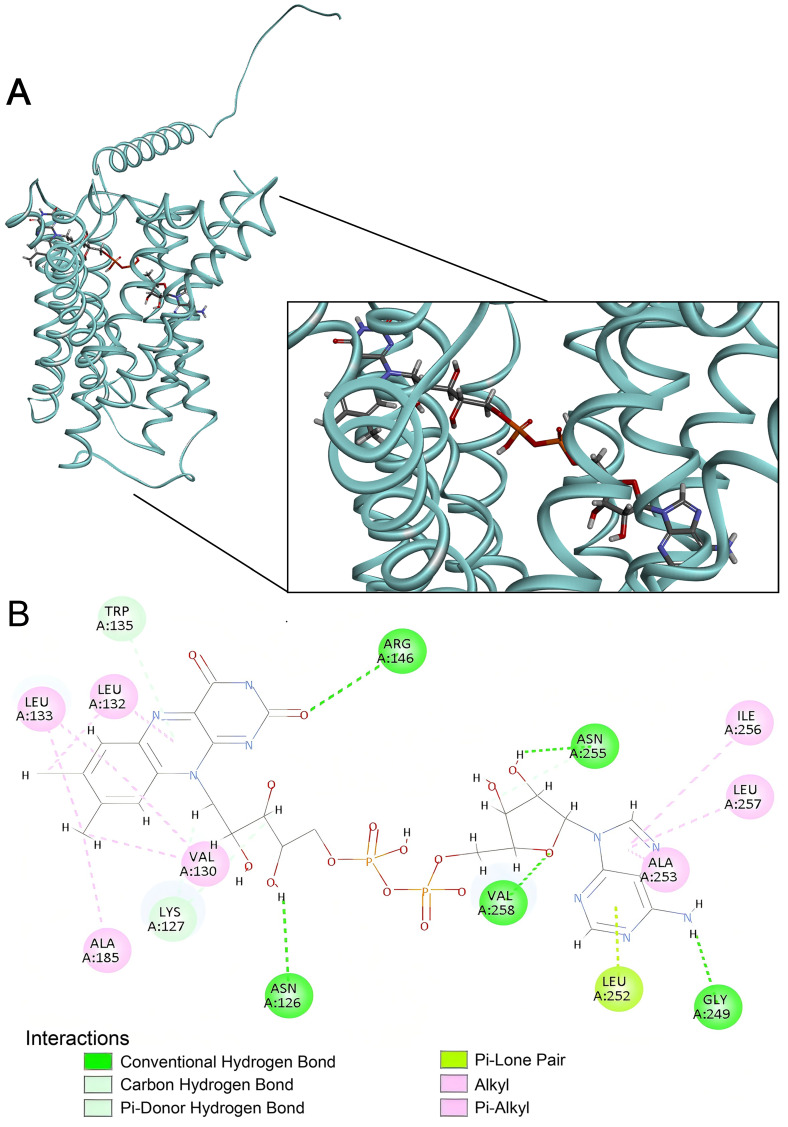
FAD docking to SLC35F4. (**A**) Representative three-dimensional visualization of the SLC35F4-FAD docking complex, with FAD shown in stick representation and the protein backbone pocket displayed as a ribbon diagram showing its predominantly α-helical secondary structure. (**B**) Two-dimensional interaction map of the SLC35F4-FAD complex using Discovery Studio 2025, highlighting residue-level contacts. Dotted lines identify predicted atomic interactions, with colors specifying interaction types as noted at the bottom of the panel.

**Figure 9 ijms-27-00512-f009:**
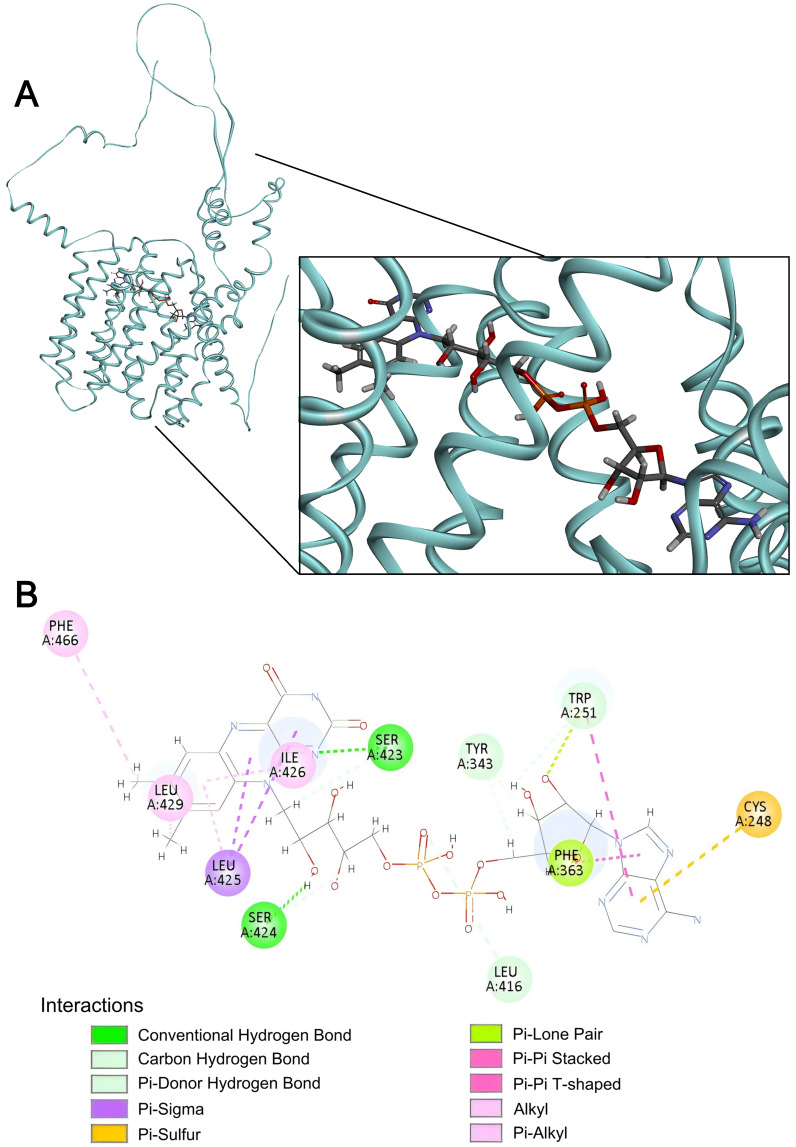
FAD docking to SLC35F5. (**A**) Representative three-dimensional visualization of the SLC35F5-FAD docking complex. (**B**) Two-dimensional interaction map of the SLC35F5-FAD complex. Panel features as presented in the Figure 8 legend.

**Figure 10 ijms-27-00512-f010:**
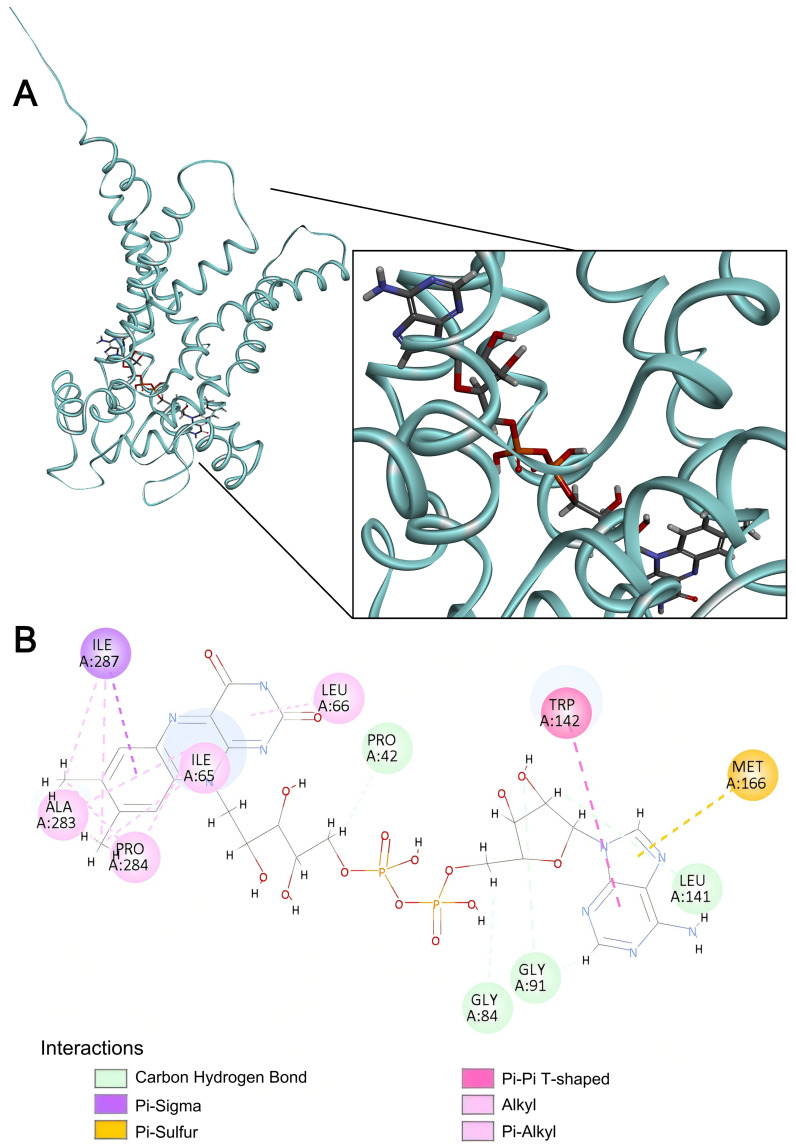
FAD docking to SLC25A32. (**A**) Representative three-dimensional visualization of the SLC25A32-FAD docking complex. (**B**) Two-dimensional interaction map of the SLC25A32-FAD complex. Panel features as presented in the Figure 8 legend.

**Table 1 ijms-27-00512-t001:** SLC35F4 and SLC35F5 subcellular localization predicted by DeepLoc v2.0.

Transporter	Subcellular Compartment	Predicted Probability	Threshold
SLC35F4	**Golgi apparatus**	**0.7088**	**0.6494**
	Cell membrane	0.4742	0.5646
	Lysosome/Vacuole	0.4302	0.5848
	Endoplasmic reticulum	0.3601	0.6090
	Peroxisome	0.1715	0.7364
	Cytoplasm	0.1148	0.4761
	Nucleus	0.1114	0.5014
	Mitochondrion	0.0865	0.6220
	Extracellular	0.0799	0.6173
	Plastid	0.0278	0.6395
SLC35F5	**Golgi apparatus**	**0.7691**	**0.6494**
	Cell membrane	0.4655	0.5646
	Lysosome/Vacuole	0.3943	0.5848
	Endoplasmic reticulum	0.3426	0.6090
	Peroxisome	0.1690	0.7364
	Cytoplasm	0.1174	0.4761
	Nucleus	0.0930	0.5014
	Mitochondrion	0.0858	0.6220
	Extracellular	0.0654	0.6173
	Plastid	0.0223	0.6395

**Table 2 ijms-27-00512-t002:** Predicted stabilizing interactions for FAD binding to SLC35F4, SLC35F5, and SLC25A32.

Interaction	SLC35F5	SLC35F4	SLC25A32
Conventional hydrogen bond	**Yes**	**Yes**	No
Carbon hydrogen bond	**Yes**	**Yes**	**Yes**
π–π stacking (parallel)	**Yes**	No	No
π–π T-shaped	**Yes**	No	**Yes**
π–σ interaction	**Yes**	No	**Yes**
π–alkyl interaction	**Yes**	**Yes**	**Yes**
Alkyl (hydrophobic) interaction	**Yes**	**Yes**	**Yes**
π–sulfur interaction	**Yes**	No	**Yes**
π–lone pair interaction	**Yes**	**Yes**	No
π–Donor Hydrogen Bond	**Yes**	**Yes**	No

## Data Availability

All relevant data are presented in the published paper or the supplied Appendix A.

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
