# Peer review of "Molecular Modeling and Gene Ontology Implicate SLC35F4 and SLC35F5 as Golgi-Associated Importers of Flavin-Adenine-Dinucleotide"

_ijms, 2026, doi:10.3390/ijms27010512_

Round 1

Reviewer 1 Report (New Reviewer)

Comments and Suggestions for Authors

Zheyun Niu and colleagues report on a possible function of SLC35F4 and SLC35F5 as FAD transporters in the Golgi apparatus/ER. For comparison the used SLC25A32, which is a mitochondrial FAD transporter. Furthermore, they question the previously proposed function of SLC35F3 as a thiamin transporter. They used exclusively in silico methods to come to these predictions.

In general, this is an interesting manuscript and the methods that have been used seem to be sound. The manuscript is principally well written and appropriate in length.

I have some points that might be considered:

  1. It is interesting to propose a function for a so far uncharacterized protein. However, it is also dangerous to relay exclusively on in silico prediction. I think that the title makes it clear that the proposed function is due to prediction and not validated by an experiment. However, I would encourage the authors to add a sentence in the Abstract and a paragraph in the Discussion to mention experimental methods (e.g. knock-out models, recombinant proteins) that can confirm such in silico predictions.

  1. Concerning the possibly overlapping function of SLC35F4 and SLC35F5. According to GTEx data SLC35F5 is generally higher expressed than SLC35F4. The expression of SLC35F5 is even higher in the cerebellum compared to SLC35F4. The preferred expression of SLC35F4 in the cerebellum does not explain an involvement in neurodegenerative disease. Since this hypothesis is in my view far fetched and not supported by any evidence, I would encourage to remove these speculations from the Discussion on page 18.

Author Response

We thank this reviewer for her/his helpful comments. Our corresponding responses in the manuscript document appear highlighted in yellow when unique to this reviewer and cyan when common to both reviewers. 

Comment 1. It is interesting to propose a function for a so far uncharacterized protein. However, it is also dangerous to relay exclusively on in silico prediction. I think that the title makes it clear that the proposed function is due to prediction and not validated by an experiment. However, I would encourage the authors to add a sentence in the Abstract and a paragraph in the Discussion to mention experimental methods (e.g. knock-out models, recombinant proteins) that can confirm such in silico predictions.

Author response. We added a sentence at the end of the Abstract and a paragraph at the end of the Discussion as recommended. 

Comment 2. Concerning the possibly overlapping function of SLC35F4 and SLC35F5. According to GTEx data SLC35F5 is generally higher expressed than SLC35F4. The expression of SLC35F5 is even higher in the cerebellum compared to SLC35F4. The preferred expression of SLC35F4 in the cerebellum does not explain an involvement in neurodegenerative disease. Since this hypothesis is in my view far fetched and not supported by any evidence, I would encourage to remove these speculations from the Discussion on page 18.

Author response. We added language in the Discussion noting the co-expression of SLC35F4 and SLC35F5 in the cerebellum and addressing the possible interplay between the two carriers. We also clarified that our discussion refers to the involvement cerebellum itself in neurological disorders (which is not controversial), and not any specific involvement of SLC35F4 or SLC35F5. We feel obliged not to delete this paragraph entirely because the preferential expression of SLC35F4 in cerebellum seems to demand at least some comment.

Reviewer 2 Report (New Reviewer)

Comments and Suggestions for Authors

The overarching goal of this work is to use gene ontology and molecular docking to identify the properties (e.g. substrate specificity, function, directionality) of the poorly understood SLC35F4 and SLC35F5. Gene ontology analysis predicted SLC35F4 and SLC35F4 to most likely localize in the Golgi apparatus. From molecular docking of 71 candidate substrates, the authors identified FAD as the likely cofactor of SLC35F4 and SLC35F5. They also correspond the molecular docking scores to binding affinities. Analysis was also compared with SLC25A32, a known FAD transporter. They conclude that SLC35F4 and SLC35F5 may import the FAD substrate in the Golgi, although the predictions have not been verified experimentally. While the current work is successful in achieving the authors’ goals, addressing the following criticisms can further strengthen their claims and the importance of this work

  1. Abstract
    1. Minor: “Ontology” is misspelled as “ontogeny” in the 6th sentence.
  2. Experimental validation
    1. In the introduction section, the authors say that there is little to no experimental evidence for certain SLC’s, motivating the need to examine orphan SLC’s. However, this work is also limited in that there is no experimental verification that confirms the predictions made from gene ontology and molecular docking. If possible, the authors should consider running biochemical assays (for example, using epitope/fluorescent tagging to detect SLC’s in organelles) or ITC / SPR to measure Kd values to supplement their work.
  3. Gene ontology
    1. As a control, has the TOPCONS approach been applied to the other members of the SLC35F subfamily? Is TOPCONS able to correctly predict the subcellular compartments of the other transporters with known functions?
  4. Molecular docking
    1. Figures 5 and 6 show comparative docking of substrates the SLC’s, however there is no explanation in the main text or supplementary information about how the docking modes are different. Are the protein receptor binding pockets different? Or does this represent the poses generated from a single docking run into a rigid protein structure?
    2. Binding affinities are backcalculated from the docking scores; these binding affinities should not be considered as absolute binding affinity since docking scores can only inform relative binding across a series of substrates and are better suited for an internal ranking. AutoDock 4 uses a semi-empirical scoring function that does not often correlate well with experimental binding affinities. The authors should discuss a limitation of using this approach to infer absolute binding affinity. Additionally, are there other reports in the literature about the binding affinity of FAD to SLC25A32? How do these values relate to the predicted affinity from docking? What temperature was used for Kd calculation?
    3. Docking of thiamine to SLC35F3 showed unfavorable binding affinity. Can the authors suggest a possible explanation to why this disagrees with the current hypothesis? What does SLC35F3 docking look like with FAD?
    4. Minor: A structure-based approach that involves AlphaFold3 structural models with gene sequence information were used to predict functional annotations. As with any structural model generated by AlphaFold3, please report the pLDDT statistics for which regions have high vs low-confidence and any additional processing steps prior to docking. Also, what is the structural homology among the three SLCs?
    5. Minor: Figures 8 to 10 show the structural basis for FAD docking to the SLC’s. For better representation, a table could be used to summarize which residue interactions are common or present across the three SLC-FAD docking complexes. 
  5. MD or MM-GBSA/MM-PBSA
    1. Docking is also limited in that substrates are placed in a rigid protein structure. Molecular dynamics simulations can further strengthen this work by showing that the ligand-residue interactions observed from docking are stable over time. Additionally, MM-GBSA or MM-PBSA can be applied to confirm the rank-ordering of the substrates docked.
  6. Conclusions
    1. Further details can be provided about specific experiments to validate the predictions in this paper. 

Author Response

We thank this reviewer for her/his insightful comments and helpful suggestions.

Cyan highlighting in the revised manuscript denotes responses to this reviewer’s comments, which we believe have strengthened the manuscript

Comment 1. Abstract. Minor:“Ontology” is misspelled as “ontogeny” in the 6th sentence.

Author response: Thank you for catching this mistake! We corrected the error in the revised manuscript.

Comment 2. Experimental validation. In the introduction section, the authors state that there is little to no experimental evidence for certain SLCs, motivating the need to examine orphan SLCs. However, this work is also limited by the lack of experimental verification confirming predictions from gene ontology and molecular docking. If possible, the authors should consider biochemical assays (e.g., epitope/fluorescent tagging to detect SLC localization in organelles) or ITC / SPR to measure Kd values to supplement their work.

Author response: The main goal of this study is to perform a comparative analysis of potential binding modes between SLC proteins and FAD based on structural models. We believe our current docking results strongly support the implication that SLC35F4 and SLC35F5 function as Golgi-associated importers of FAD, and acknowledge that this hypothesis requires further experimental validation that is beyond the scope of this computational study. We have now added a paragraph in the Discussion elaborating on the limitations of our analyses and suggesting appropriate methods for future experiments.

Comment 3. Gene ontology. As a control, has the TOPCONS approach been applied to other members of the SLC35F subfamily? Is TOPCONS able to correctly predict the subcellular compartments of other transporters with known functions?

Author response: TOPCONS is a well-established and widely used tool for predicting transmembrane topology that has been used for many years in studies of solute carrier (SLC) proteins and other multi-pass membrane transporters. TOPCONS is commonly applied to predict the number of transmembrane helices, membrane orientation, and overall topology consistency. Nevertheless, we did not find studies that systematically applied TOPCONS specifically to members of the SLC35F subfamily. However, several studies on other members of the SLC35 family have combined sequence analysis with topology prediction to characterize transmembrane features, which were later supported by functional experiments. For example, in the study “SLC35G3 is a UDP-N-acetylglucosamine transporter for sperm glycoprotein formation and underpins male fertility in mice”, the authors performed a detailed structural and functional analysis of the Golgi-associated transporter SLC35G3. Its multi-pass membrane structure and topology were predicted using AlphaFold2 and TOPCONS. Similarly, in “Conserved Glu-47 and Lys-50 residues are critical for UDP-N-acetylglucosamine/UMP antiport activity of the mouse Golgi-associated transporter Slc35a3”, the authors used TOPCONS together with MEMSAT-SVM, HMMTOP, and TMHMM to identify conserved residues near transmembrane regions. Importantly, TOPCONS is designed to predict transmembrane topology, not subcellular localization. Therefore, TOPCONS cannot directly predict whether a protein is located in the endoplasmic reticulum, Golgi apparatus, mitochondria, or plasma membrane.

In this study, subcellular localization was predicted using DeepLoc 2.1, a tool specifically developed for this purpose. DeepLoc 2.1 is a deep learning–based method trained on proteins with experimentally known localization. Its predictions are consistent with transporters of known function and show better performance than earlier membrane localization models such as Mem-ADSVM. As a validation, DeepLoc 2.1 correctly predicted the known mitochondrial localization of the FAD transporter SLC25A32 in our analysis.

Comment 4a. Molecular docking. Figures 5 and 6 show comparative docking of substrates to the SLCs; however, there is no explanation in the main text or supplementary information about how the docking modes differ. Are the protein receptor binding pockets different, or do these represent poses generated from a single docking run into a rigid protein structure?

Author response: The different docking modes shown in Figures 5 and 6 mainly come from multiple ligand conformations (modes or poses) generated from a single AutoDock Vina docking run under the assumption of a rigid protein receptor. These modes do not represent different binding pockets. In this study, the protein receptor was kept rigid during docking, and only ligand flexibility was allowed. Therefore, the different docking modes reflect differences in ligand orientation and conformation within the same predicted binding cavity.

To test the robustness and reproducibility of the docking results, we performed multiple independent docking runs using the same receptor structure and identical docking parameters. Across these independent runs, some variation in ligand conformation was observed. However, the top-ranked docking poses consistently localized to the same predicted binding region, and their predicted binding affinity values remained stable across different runs. These results indicate that the docking modes shown in the figures are not artifacts of a single random docking run, but represent reproducible local energy minima within a rigid-receptor docking framework.

We also acknowledge that rigid docking methods cannot capture protein flexibility or induced-fit effects, which is an inherent limitation of the docking approach used in this study. Accordingly, we added relevant language in a new Discussion paragraph (as noted in our response to comment 2a above), where we recognize the utility of molecular dynamics analyses to overcome this limitation. In future work, we plan to validate the proposed interaction through experimental approaches, such as FAD transport assays, ITC or SPR measurements to determine binding affinities (Kd values), and mutational analysis of key residues within the predicted binding pocket.

Comment 4b. Binding affinities are back-calculated from docking scores; these should not be considered absolute binding affinities since docking scores only inform relative binding across a series of substrates and are better suited for internal ranking. AutoDock4 uses a semi-empirical scoring function that often does not correlate well with experimental binding affinities. The authors should discuss this limitation. Additionally, are there reports in the literature about the binding affinity of FAD to SLC25A32? How do these values relate to the predicted affinity from docking? What temperature was used for Kd calculation?

Author response: We agree with the reviewer that binding affinities calculated from docking scores should not be taken as absolute binding affinities. In this study, we used AutoDock Vina, not AutoDock4. AutoDock Vina uses a semi-empirical scoring function that is mainly designed for relative ranking of ligands under the same computational conditions, rather than for accurate prediction of binding free energy or dissociation constants (KD) under experimental conditions. Therefore, we used docking scores only for relative comparisons between different substrates or different proteins, and not for estimation of absolute binding affinity. In addition, the scoring function in AutoDock Vina does not explicitly include a temperature parameter. The docking calculation is not based on thermodynamic equilibrium or molecular dynamics simulation, but on empirical evaluation of ligand–receptor interactions under default standard settings. As a result, no temperature parameter was set in our docking analysis. From our perspective, the important point is that the predicted affinity of the known FAD transporter SLC25A32, back-calculated at 298 K, lies between those of SLC35F4 and SLC35F5, and KD’s of all three carriers lie well below our best estimates of intracellular FAD concentration.

Regarding the experimental binding affinity between FAD and SLC25A32, we found no published studies that directly measured the dissociation constant (KD) for this interaction. SLC25A32 is a mitochondrial inner membrane transporter, and its functional link to FAD has mainly been established through transport and functional rescue experiments rather than direct binding assays. For example, Peng et al., in “Mitochondrial FAD shortage in SLC25A32 deficiency affects folate-mediated one-carbon metabolism”, used SLC25A32 knockout and knock-in mouse models and showed that loss of SLC25A32 specifically blocks mitochondrial FAD uptake and leads to mitochondrial FAD deficiency.

In this context, the docking results in our study should be viewed as relative comparisons across different proteins. For example, we observed that the predicted affinity of FAD for SLC35F4 and SLC35F5 is closer to that of the known mitochondrial FAD transporter SLC25A32, whereas the predicted affinity of FAD for SLC35F3 is lower. Based on this relative comparison, we propose that SLC35F4 and SLC35F5 may function as FAD transporters in the Golgi apparatus.

Comment 4c. Docking of thiamine to SLC35F3 showed unfavorable binding affinity. Can the authors suggest a possible explanation for why this disagrees with the current hypothesis? What does SLC35F3 docking look like with FAD?

Author response: Honestly, our main explanation for the discrepancy between our SLC35F3 results and the current thiamine transporter hypothesis is that the latter is simply wrong, for several reasons. The thiamine studies don’t account for the predominant intracellular localization of SLC35 family members. The investigators picked thiamine as a candidate substrate based on dubious amino acid sequence comparisons. They measured thiamine uptake by bacteria, and the experiments lacked important specificity controls, including comparison to uptake of different substrates. And the thiamine transport hypothesis doesn’t account for needed antiport substrates. We recognize that solid experimental data generally override contradictory computational analyses, but in this instance the experiments lack the rigor necessary to be confident that SLC35F3 is a thiamine transporter.

Comment 4d. Minor: A structure-based approach involving AlphaFold3 structural models with gene sequence information was used to predict functional annotations. As with any AlphaFold-generated structural model, please report pLDDT statistics indicating high- vs low-confidence regions and any additional preprocessing steps prior to docking. Also, what is the structural homology among the three SLCs?

Author response: We now refer to the pLDDT statistics from the AlphaFold3 models in the manuscript Results text, and have used them to mark high- and low-confidence regions accordingly in the predicted structures presented in Figure S4 of Supplementary Materials.

Regarding additional pre-processing steps, before docking we prepared both protein and ligand structures using AutoDock tools. For the protein structures, hydrogen atoms were added and water molecules removed. For the small-molecule ligands, hydrogen atoms were added and rotatable bonds were defined. After structure preparation, the docking pocket coordinates were defined. Docking poses were then evaluated by comparing docking scores, and the relative optimal binding conformations were obtained based on these scores.

To assess structural homology among the proteins, we performed structural alignment using TM-align (Y. Zhang and J. Skolnick, TM-align: A protein structure alignment algorithm based on TM-score, Nucleic Acids Research, 33:2302–2309). The results showed that F3 and F4 have a TM-score of 0.69, which is comfortably above the 0.5 threshold used to define the same protein fold. This result indicates that F3 and F4 share a highly conserved overall structural framework. A total of 384 residues were successfully aligned, with approximately 54% sequence identity, and an RMSD of 2.75 Å. This result suggests that the core transmembrane regions are highly similar, while structural differences are mainly located in non-transmembrane loop regions. The TM-score between F3 and F5 was 0.55, which is also above the fold threshold. In this comparison, 322 residues were aligned. Although the sequence identity in the aligned regions was about 20%, the RMSD was 3.47 Å. Similarly, the TM-score between F4 and F5 was approximately 0.55, again above the 0.5 threshold. In this case, 321 residues were aligned, with about 20% sequence identity and an RMSD of 3.49 Å.

Comment 4e. Minor: Figures 8–10 show the structural basis for FAD docking to the SLCs. For better representation, a table could be used to summarize which residue interactions are common or present across the three SLC–FAD docking complexes.

Author response: To improve clarity and highlight the common and recurrent interactions from the docking results, we have now summarized the interacting residues observed for the three SLC–FAD docking complexes in Table 2 of the revised manuscript.

Comment 5. MD or MM-GBSA/MM-PBSA. Docking is limited in that substrates are placed in a rigid protein structure. Molecular dynamics simulations can further strengthen this work by showing that ligand–residue interactions observed from docking are stable over time. Additionally, MM-GBSA or MM-PBSA can be applied to confirm the rank-ordering of the substrates docked.

Author response: We fully agree that molecular dynamics (MD) simulations have clear advantages for studying ligand–protein interactions. In particular, MD can help evaluate the stability of ligand–residue interactions over time. Methods such as MM-GBSA or MM-PBSA can also be used to re-evaluate docking results at the energy level. Therefore, MD can provide useful complementary information to docking analysis.

However, it should be noted that MD simulations are still computational prediction methods based on force fields and initial structural assumptions. The results strongly depend on parameter settings, simulation time, and force-field choice. As a result, MD simulations cannot fully replace real biophysical or biochemical experiments. In the absence of experimental binding data for validation, MD mainly provides theoretical support for possible interaction modes rather than direct experimental evidence.

Given that the main goal of this study is to perform a comparative analysis of potential binding modes between SLC proteins and FAD based on structural models, we believe that the current docking results are sufficient to generate the hypothesis that SLC35F4 and SLC35F5 may function as FAD transporters in the Golgi apparatus. We have also explicitly stated limitation of docking analysis in the revised manuscript, recognized the utility of MD for potentially overcoming it as noted in responses to comments above, and acknowledged the need for experimental validation of our findings in future work.

Comment 6. Conclusions. Further details can be provided about specific experiments to validate the predictions in this paper.

Reply: As noted above, in the new Discussion paragraph (on manuscript pages 17-18) we have explicitly stated the limitation of docking analysis, and described several specific experimental strategies for validating the predictions in our study.

Round 2

Reviewer 2 Report (New Reviewer)

Comments and Suggestions for Authors

Thank you to the authors for carefully considering the criticisms. The authors have responded point-by-point, adding to their discussion in the manuscript. Although additional experimental work may still be necessary to verify the findings, there is good basis from their approach that SLC35F4 and SLC35F5 can bind and import FAD.

This manuscript is a resubmission of an earlier submission. The following is a list of the peer review reports and author responses from that submission.

Round 1

Reviewer 1 Report

Comments and Suggestions for Authors

The authors present a computational analysis of the human SLC35F3/4/5 proteins with the aim of elucidating details about their function. Phylogenetic analysis corroborated previous findings that the SLC35F3/4/5 proteins form a distinct clade from their peers in the SLC35F subgroup. Automatic functional annotation suggested localization in the Golgi, typical for SLC35 proteins. The same annotation pipeline also suggested transmembrane transport activity, which also seems plausible, if not very surprising, for a member of a transmembrane transporter family.

The authors furthermore docked a panel of 71 molecules in silico into the assumed binding sites of these transporters and analyzed predicted protein-ligand interactions. Here, however, the structures and images presented in Figures 7, 8, and 9 raise concerns about the stereochemistry of the docked ligands. The interaction diagrams from Discovery Studio in all 3 cases show positive charges on several nitrogen atoms in both the flavin and adenine groups, indicating "unfavorable positive-positive" interactions with nearby lysine headgroups, even though neither of these groups should contain positively charged nitrogens. Furthermode, the 3D images of the docked ligands in these figures show that neither the flavin nor the adenine groups show the expected planar geometry in their aromatic ring systems. Since the interaction diagrams lack any double bonds in the aromatic rings, I suspect that those rings have been incorrectly recognized and treated as cyclohexyl (non-aromatic non-planar) rings, which is erroneous. The non-planarity of the 3D docked structures indicate that the incorrect rings are likely to have been present already during the docking process. The issue is concerning because without the aromatic rings, the authors might have docked a different ligand with a different number of hydrogens and atom types as they would have wished. This might affect overall docking results and conclusions from them.

In order to assess the outcome of the docking results, which is one of the main messages of the manuscript, these and other potential issues with stereochemistry must be fixed. The authors do not present docked structures of other ligand in the main figures, but they are advised to check all docked ligands for similar errors. In addition, FAD and other similar ligands have multiple stereocenters, the 3D geometry of which must also be double-checked to make sure they correspond to biologically relevant stereoisomers. After these fixes in ligand geometry, it is also advisable that the authors double-check whether the correct atom types are assigned to ligand atoms by the docking software before they repeat the docking studies. The repeated docking studies with the correct ligand geometries will likely lead to different docking scores and docked poses, so the outcome will have to be re-evaluated in light of this.

If the repeated docking studies still support the original hypothesis of the authors, the manuscript can be a valuable addition to the deorphanization efforts of SLC transporters. On the other hand, in that case, one might be concerned that the stereochemistry of the ligand does not play a crucial role in its recognition.

Reviewer 2 Report

Comments and Suggestions for Authors

The subject of the manuscript predicting possible Golgi FAD transporters (SLC35F4 and SLC35F5) is potentially interesting, but the overall quality of the work does not meet publication standards. The title itself lacks clarity and precision, showing typographical errors (“SLC35F4 and SLC35F4”) and failing to reflect the true scope of the study. A well-structured and informative title is essential for communicating the main contribution of the paper, but here it gives a confusing first impression. In addition, the logical flow of the manuscript is weak: the sections are not well connected, transitions between results are abrupt, and the reasoning from prediction to biological conclusion is not coherent. The conclusion is poorly written, repeating general background rather than summarizing the findings or explaining their significance. The figures are of low quality, with unclear labeling and poor resolution, which makes interpretation difficult. The language throughout is inconsistent with grammatical errors. Overall, the manuscript requires a complete reorganization in structure, title, and writing, along with stronger scientific evidence to support its claims. Therefore, I recommend rejection of the manuscript in its current form. 

Author Response

We have extensively revised the manuscript Title, Results (with additional data and revised figures), Discussion, and Conclusions per the reviewer's comments.

Round 2

Reviewer 1 Report

Comments and Suggestions for Authors

Author Response

We appreciate the reviewer’s concerns about the apparent clashing of the FAD ligand with the the SLC35F4 alpha helices depicted in Figure 8 panel B, which we had also noticed prior to the first resubmission. This phenomenon is fairly common in renderings of docked ligands visualized in Vina, a fact that can easily be confirmed by online search for questions about and solutions to such problems. Indeed, we can provide numerous examples of such renderings in the published literature if the reviewer needs further confirmation. Most importantly, as noted in our response to initial review regarding the issues with the FAD structure (which we resolved by upgrading to AutoDock Vina 2025), visualization problems are not indicative of flawed docking results (or of post hoc manipulation, as the reviewer accused). We used the definitive structures of all 71 ligands as retrieved from PubChem in SDF format and directly converted to PDBQT for docking, without alteration. The reviewer is correct in perceiving our reluctance to repeat docking experiments when there is nothing wrong with them.

To address the reviewer’s concerns with the visualizations, and per the explanations described below in Technical Details, we re-visualized the docking results for the highest affinity FAD interactions with each of the carriers (SLC35F4, SLC35F5, and SLC25A32; all Mode 1). These renderings are now presented in revised Figures 8–10. Specifically, we adjusted the rendering settings to decrease the width of the ribbons and provide a more accurate depiction of the backbone polypeptide chain, which is the recommended “fix” for apparent (and artifactual) penetration of stick figure ligands through protein secondary structure, as well as for apparent breaks in the ligand structure.

Technical Details

In the visualization of docking results, the phenomenon in which a ligand appears to pass through an α-helix is actually a rendering artifact, rather than an error in the structure or docking calculation. In molecular visualization software such as BIOVIA Discovery Studio, PyMOL, and VMD, displaying proteins in cartoon (ribbon) mode provides a geometric simplification of secondary structure rather than a true atomic surface. The ribbon is a smoothed geometric envelope interpolated from the backbone atoms, and depending on settings (for example, a wider or thicker ribbon) it may visually overlap or occlude stick-style ligands, especially when the ligand lies within the helical groove or close to the main chain, thereby creating the illusion that the ligand passes through the helix.

In addition, molecular visualization programs employ clipping planes and depth-sorting algorithms to control the display range and spatial ordering. When an object lies near a clipping boundary, or a depth-sorting conflict occurs, parts of the image may be cut or obscured, further enhancing the illusion of the ligand penetrating the helix. This effect is purely visual and does not influence docking calculations or scoring results. Docking programs such as AutoDock Vina perform scoring and conformational searches entirely based on atomic coordinates and energy functions, independent of any subsequent visualization settings. Adjusting the ribbon thickness or transparency (e.g., 40–60%), or replacing the ribbon representation with a semi-transparent molecular surface, can substantially reduce the overlap; however, due to the intrinsic limitations of the cartoon representation, the effect cannot be completely eliminated.
